# Optimizing the Positioning of Soil Moisture Monitoring Sensors in Winter Wheat Fields

**Xiaojun Shen [1,2], Jing Liang [3], Ketema Tilahun Zeleke [2,4]**  **, Yueping Liang [1], Guangshuai Wang [1], Aiwang Duan [1], Zhaorong Mi [1], Huifeng Ning [1], Yang Gao [1,\*] and Jiyang Zhang [1,\*]**

[1] Key Laboratory of Crop Water Use and Regulation, Ministry of Agriculture, Farmland Irrigation Research Institute, Chinese Academy of Agricultural Sciences, Xinxiang 453002, Henan, China; xjshen2016@gmail.com (X.S.); yueping0520@163.com (Y.L.); wangguangshuai@caas.cn (G.W.); duanaiwang@caas.cn (A.D.); mizhaorong@caas.cn (Z.M.); ninghuifeng@caas.cn (H.N.)

[2] Graham Centre for Agricultural Innovation an Alliance between NSW Department of Primary Industries and Charles Sturt University, Wagga Wagga NSW 2650, Australia; kzeleke@csu.edu.au

[3] Department of Environmental Sciences, University of California Riverside, Riverside, CA 92507, USA; jlian014@ucr.edu

[4] School of Agricultural & Wine Sciences, Charles Sturt University, Wagga Wagga NSW 2650, Australia

\* Correspondence: yanggao.firi@gmail.com (Y.G.); firizjy@163.com (J.Z.); Tel.: +86-373-339-3384 (Y.G.); +86-373-339-3384 (J.Z.)

**Abstract:** Collecting accurate real-time soil moisture data in crop root zones is the foundation of automated precision irrigation systems. Soil moisture sensors (SMSs) have been used to monitor soil water content (SWC) in crop fields for a long time; however, there is no generally accepted guideline for determining optimal number and placement of soil moisture sensors in the soil profile. In order to study adequate positioning for the installation of soil moisture sensors in the soil profile, six years of field experiments were carried out in North China Plain (NCP). Soil water content was measured using the gravimetric method every 7 to 10 days during six growing seasons of winter wheat (*Triticum aestivum* L), and root distribution was measured using a soil core method during the key periods of winter wheat growth. The results from the experimental data analysis show that SWC at different depths had a high linear correlation. In addition, the values of correlation coefficients decreased with increasing soil depth; the coefficient of variation (CV) of SWC was higher in the surface layers than in the deeper layers (depths were 0–40 cm, 0–60 cm, and 0–100 cm during the early, middle, and last stages of winter wheat, respectively); wheat roots were mainly distributed in the surface layer. According to an analysis of CV for SWC and root distribution, the depths of planned wetted layers were determined to be 0–40 cm, 0–60 cm, and 0–100 cm during the sowing to reviving stages (the early stage of winter wheat), returning green and jointing stages (the middle stage of winter wheat), and heading to maturity stage (the last stage of winter wheat), respectively. The correlation and R-cluster analyses of SWC at different layers in the soil profile showed that SMSs should be installed 10 and 30 cm below the soil surface during the winter wheat growing season. The linear regression model can be built using SWC at depths of 10 and 30 cm to predict total average SWC in the soil profile. The results of validation showed that the developed model provided reliable estimates of total average SWC in the planned wetted layer. In brief, this study suggests that suitable positioning of soil moisture sensors is at depths of 10 and 30 cm below the soil surface.

**Keywords:** soil water content; soil profile; soil moisture sensor; winter wheat; correlation analysis

## 1. Introduction

In the North China Plain (NCP), rich resources of light and heat have created favorable conditions for crops. The NCP supplies approximately 61% of the wheat and 45% of the maize in China, and has become one of the country's most important high-quality wheat-producing regions [1]. Due to the impact of monsoon season, when approximately 25 to 40% of precipitation falls (annual rainfall: 400–800 mm) during the winter wheat growing stage [2], soil water storage is the primary source of water for winter wheat grow. Irrigation, precipitation, plant evapotranspiration, and soil evaporation are the main factors that affect soil water storage in the winter wheat field, irrigation is one of the most effective ways to increase yield in NCP. However, the irrigation water shortage has become one of the main restrictive factors in producing high quality and yield for winter wheat. Precision irrigation technology can potentially improve water use efficiency by better targeting water to precisely meet the demands of crops on a local basis without wasting time and water. In addition, growers can save money on labor by letting real-time sensors measure soil properties. The economic and environmental benefits of taking account of precision agriculture can contribute to the long-term sustainability of production agriculture.

Fast and accurate monitoring of soil moisture is the basis for precision irrigation, which provides support for decision-making and irrigation automation [3,4]. Soil moisture sensors have been used to monitor soil water content in the field for a long time [5,6]. More and more research has reported that soil moisture sensors can measure soil water content rapidly and precisely, and study the effects of different types of SMS [7–10]. The number and locations of sensors will directly impact the accuracy of soil moisture measurements in the soil profile [11,12]. Reducing the number of sensors is one of the most efficient ways to decrease the cost of a precision irrigation system, while a smaller number of sensors, with the proper placement, can represent the average soil moisture across the soil profile. There are some differences between the soil moisture at the root zone and the average soil moisture in the soil profile because of differences of root distribution in the soil profile at different growing stages. There is high linear correlation among soil water content (SWC) at different depths in the soil profile. Therefore, we can use the SWC of one layer to indicate the SWC near the adjoining layers in order to reduce the number of sensors. Several studies have been conducted to investigate the proper positioning of soil moisture sensors [13]. Pogue and Pooley indicated that proper positioning of SMSs in the soil can be determined by the empirical function [14]. Haise and Hagan recommended that SMSs should be buried at the upper and lower limits of root water absorption [15], or two SMSs should be buried at different depths in the root zone, one twice as deep as the other [16]. Yang et al. recommended that 10, 20, and 50 cm below the surface are proper positions for SMSs based on clustering methodology for the different layers of SWC in the soil profile [13]. Gao et al. determined the number and depths of SMSs during different growing stages based on the relationship of different layers of soil water content in the soil profile [17]. Machado C. and Coelho recommended that soil moisture sensors should be placed at distances of 20% to 75% of the canopy radius from plants and at depths of 0–20 cm, 20–40 cm, and 40–60 cm [18].

There are several reported studies on the positioning of SMSs. However, most of these studies have focused on the statistical analysis for SWC values at different layers in the soil profile during the crop growing season, rarely considering water uptake and root distribution. There are significant differences in soil water storage efficiency between the different depths at the same stage for the winter wheat, because crop roots only uptake water from the root zone. At the same time, soil water storage efficiency at the same depth is not same at the different stage of the crop growth, because the root distribution does not remain constant at different growing stages, especially for annual plants such as winter wheat. In addition, soil water storage efficiency at different soil depths varies throughout the growing season, since the water-absorbing capacity and root distribution of winter wheat are different at different growing stages. Therefore, the overall objective of this study was to improve the management of precise irrigation and automated control for winter wheat fields in the NCP. More specifically, this study aims to (i) determine suitable depths of wetting layers for the winter

wheat based on the temporal and spatial variations of soil moisture and the root distribution in the soil profile, because the soil moisture in the root zone is very important to the irrigation index determine; (ii) optimize the placement of SMSs in the soil profile based on the relationship between average SWC in the soil wetting layers (the depth of soil layer that is determined by the depth of root distribution) and SWC at different layers in the soil profile; and (iii) determine the quantitative relationship between farmland soil moisture in planned wetting layers and soil water content measured by SMSs at the proper positions. This study will provide valuable information to determine the optimum numbers and positioning of SMSs in order to optimize the irrigation index for precise irrigation with automation control based on soil moisture monitoring for winter wheat and other crops in the NCP and elsewhere.

## 2. Materials and Methods

### 2.1. Experimental Site

The field experiments were carried out in a maize/wheat rotational field at Xinxiang City ($35°19'$ N, $113°53'$ E, elevation 73.2 m) and Qinyang City ($35°04'$ N, $112°55'$ E, elevation 150 m) of Henan Province, in the southern part of the North China Plain, during 1998–1999, 2000–2002, 2003–2005, 2007–2008, 2011–2012, and 2013–2014 (Figure 1). The SMSs were installed in a 120 cm thick soil profile in both experimental sites to monitor SWC. At the experimental site in Xinxiang, average annual rainfall and temperature during the winter wheat growing season are 165 mm and 9.9 °C, respectively. The soil texture is sandy loam with a mean bulk density of 1.35 g·cm$^{-3}$ and mean field capacity of 32.4% (volumetric basis). Average available N (nitrogen), P (phosphorus), and K (potassium) contents of the top layer of soil (0–20 cm) are 59.28, 11.97, and 123.54 mg·kg$^{-1}$, respectively. Soil organic matter content is 1.64 g·kg$^{-1}$ and soil pH is 8.6. At the experimental site in Qinyang, average annual rainfall and temperature are 593.5 mm and 14.5 °C, respectively. The soil texture is loam with a mean bulk density of 1.44 g·cm$^{-3}$ and mean field capacity of 37.4% (volumetric basis). Average available N, P, and K contents of the top layer of soil (0–30 cm) are 86.00, 11.10, and 62.80 mg·kg$^{-1}$, respectively. Soil organic matter content is 8.80 g·kg$^{-1}$.

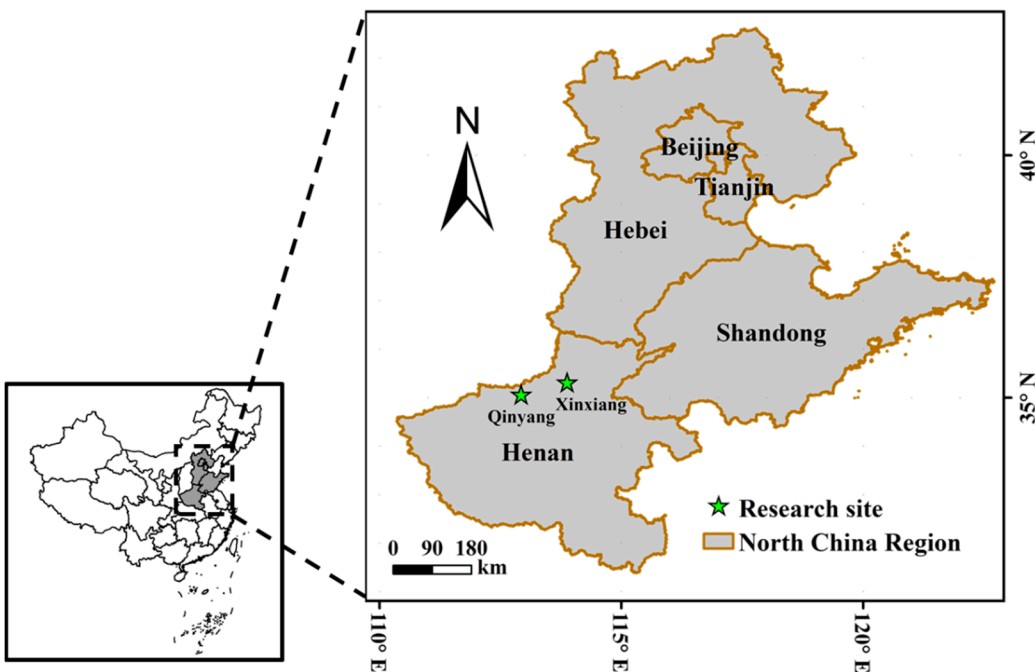

**Figure 1.** Location of North China Plain and the field experiment site. E: East longitude; N: North latitude.

A popular winter wheat variety was sown at the rate of 180 kg·ha$^{-1}$ in the middle of October each year. Each plot was 3 m wide and 30 m long, the row space was 20 cm (15 rows in each plot), planted in

a north-south direction. In all treatments, nitrogen–phosphorus–potassium (NPK) compound fertilizer was applied at rate of 750 kg·ha$^{-1}$ (N: 135.0 kg·ha$^{-1}$; $P_2O_5$: 135.0 kg·ha$^{-1}$; and $K_2O$: 37.5 kg·ha$^{-1}$) before sowing, while granular urea at a rate of 300 kg·ha$^{-1}$ (N: 138.0 kg·ha$^{-1}$) was manually broadcast at the regreening and jointing stages of winter wheat. Irrigation was applied using a border irrigation system, and treatments were replicated three times. The applied water in each plot was measured using a mechanical flow meter.

*2.2. Measurements*

Soil water content (SWC) was measured every 7 to 10 days by the gravimetric method [19] during the growing seasons for the winter wheat (we always measured every 10 days during the early stage for the winter wheat, because these stage the temperature is low and the plant is small, the evapotranspiration in field is small; during the middle stage and last stage we usually measured every seven days, because the temperature is high and the plant growth demand more water). The soil samples were collected at 20 cm increments down to 120 cm in every treatment; one soil sample was collected for each irrigation treatment.

Soil water content was measured every day by soil moisture sensors (SWR-3), three repetitions were collected for the full irrigation treatment, the sensors were installed 5, 15, 25, 35, 45, 55, 65, 75, 85, and 95 cm from the surface, and the SMSs were produced by the Beijing Zhihai Electronic Instrument Co. Ltd (Beijing, China).

To study the morphology of the root system in the soil profile, soil cores were taken according to the method of Jha et al. [20] and Böhm [21], using a 10 cm diameter corer before the overwintering stage, returning green stage, jointing stage, flowering stage, and grain filling stage. Two root samples were collected at 10 cm increments down to 1.2 m in the treatment of full irrigation. Root samples were taken to the laboratory and the soil was manually removed in the washing cans. The resulting mixture of roots and organic debris was then placed in a polythene bag and refrigerated until it could be sorted. After separating the roots from other debris, root length was measured based on the line-intersect method using a 1.27 cm grid [22]. The value of root length density can be calculated by Equation (1):

$$RLD = \frac{L_R}{V_R} \qquad\qquad (1)$$

where *RLD* is root length density (cm·cm$^{-3}$), $L_R$ is total root length of the sample (cm), and $V_R$ is total volume of the sample (cm$^{-3}$).

## 3. Results

*3.1. Suitable Depth of Soil Moisture Monitoring*

Field soil moisture variation in the root zone is influenced by precipitation, irrigation, soil evaporation, and root water uptake by crops. Figure 2 shows variations of soil water content at different layers in the soil profile during the growing season in 2007 to 2008. We can observe high fluctuations of SWC at the surface soil layers and slight changes of SWC at the deep soil layers during the growing season. Additionally, the variation of SWC decreases with increased depth in the soil profile.

The average value (AV), standard deviation (SD), and coefficient of variation (CV) of SWC at different layers during five growing stages are shown in Table 1. The CV of SWC is largest in the surface layers during different growing stages of winter wheat and the value decreases with increasing depth in the soil profile. The CV of SWC is relatively higher at 0–40 cm, 0–60 cm, 0–100 cm, and 0–100 cm than at 40–120 cm, 60–120 cm, 100–120 cm, and 100–120 cm during the early, the middle, the last stages and the whole growing season, respectively.

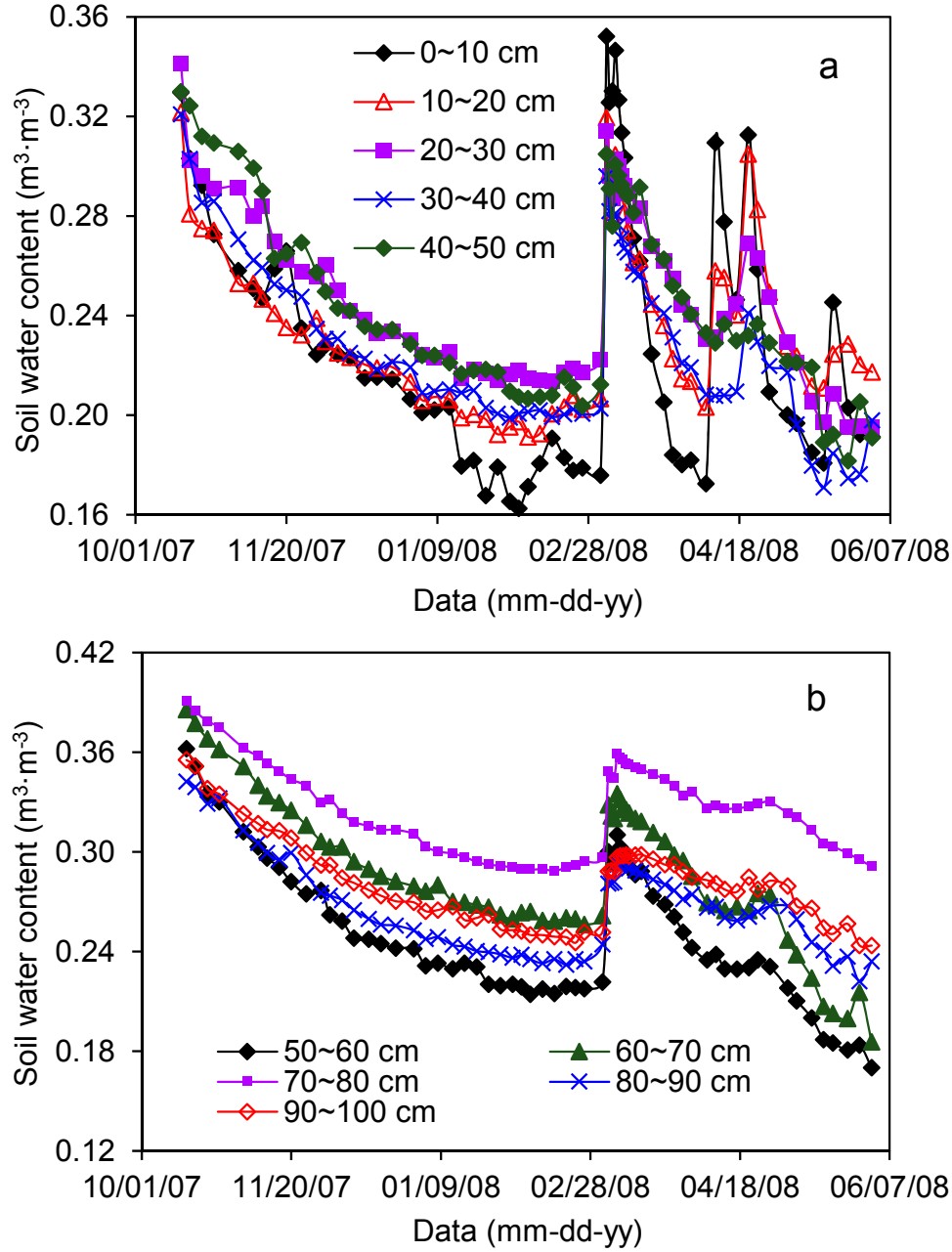

**Figure 2.** Variations of soil moisture at different layers in the soil profile during the growing season of winter wheat in 2007–2008.

To determine the proper installation depth of SMSs during different growing stages of winter wheat, the index of growth and development of roots was investigated at the key growing stages between 2011 and 2012. This study used the concept of effective root depth, which is the depth of soil that contains 80% of total root length [23]. Figure 3 shows variations of root length density at different layers in the soil profile during the growing season. We can observe that root length density decreases with increasing depth of layers in the soil profile: 97.85% of total root length was observed in the top layers 0–40 cm from the surface during the sowing to reviving stages, 96.22% was observed in the top layer of 0–40 cm during the returning green stage, 93.45% was observed in the top layer of 0–60 cm during the jointing stage, 95.12% was observed in the top layer of 0–60 cm during the flowering stage, and 98.21% was observed in the top layer of 0–100 cm during the grain filling stage.

**Table 1.** Average value (AV), standard deviation (SD), and coefficient of variation (CV) of soil water content at different layers in the soil profile during different growing stages of winter wheat.

| Depth (cm) | Early Stage | | | Middle Stage | | | Last Stage | | | Whole Growing Season | | |
|---|---|---|---|---|---|---|---|---|---|---|---|---|
| | AV (cm³·cm⁻³) | SD (cm³·cm⁻³) | CV (%) | AV (cm³·cm⁻³) | SD (cm³·cm⁻³) | CV (%) | AV (cm³·cm⁻³) | SD (cm³·cm⁻³) | CV (%) | AV (cm³·cm⁻³) | SD (cm³·cm⁻³) | CV (%) |
| 1998–1999 | n = 120 | | | n = 96 | | | n = 84 | | | n = 300 | | |
| 0–20 | 0.226 | 0.053 | 23.6 | 0.224 | 0.067 | 29.96 | 0.146 | 0.06 | 41.25 | 0.203 | 0.071 | 34.93 |
| 20–40 | 0.239 | 0.034 | 14.49 | 0.221 | 0.043 | 19.74 | 0.162 | 0.051 | 31.82 | 0.212 | 0.054 | 25.5 |
| 40–60 | 0.263 | 0.023 | 8.79 | 0.258 | 0.023 | 8.99 | 0.197 | 0.04 | 20.87 | 0.243 | 0.042 | 17.47 |
| 60–80 | 0.271 | 0.018 | 6.7 | 0.271 | 0.016 | 5.82 | 0.214 | 0.034 | 16.2 | 0.254 | 0.036 | 14.21 |
| 80–100 | 0.275 | 0.015 | 5.67 | 0.28 | 0.01 | 3.48 | 0.238 | 0.027 | 11.47 | 0.266 | 0.027 | 10.26 |
| 100–120 | 0.28 | 0.013 | 4.75 | 0.286 | 0.007 | 2.42 | 0.258 | 0.019 | 7.52 | 0.275 | 0.02 | 7.17 |
| 2000–2001 | n = 168 | | | n = 168 | | | n = 96 | | | n = 432 | | |
| 0–20 | 0.289 | 0.029 | 10.21 | 0.237 | 0.064 | 27.77 | 0.187 | 0.069 | 36.41 | 0.248 | 0.07 | 28.8 |
| 20–40 | 0.259 | 0.02 | 7.83 | 0.232 | 0.038 | 16.91 | 0.194 | 0.049 | 25.69 | 0.235 | 0.047 | 20.04 |
| 40–60 | 0.265 | 0.019 | 7.19 | 0.24 | 0.024 | 10.39 | 0.219 | 0.038 | 17.43 | 0.246 | 0.035 | 14.42 |
| 60–80 | 0.268 | 0.021 | 7.65 | 0.249 | 0.017 | 7.07 | 0.237 | 0.028 | 12.17 | 0.255 | 0.028 | 11.3 |
| 80–100 | 0.272 | 0.02 | 7.32 | 0.258 | 0.012 | 4.88 | 0.257 | 0.023 | 9.07 | 0.264 | 0.022 | 8.46 |
| 100–120 | 0.278 | 0.019 | 6.93 | 0.266 | 0.014 | 5.1 | 0.273 | 0.024 | 8.7 | 0.274 | 0.022 | 7.97 |
| 2001–2002 | n = 900 | | | n = 360 | | | n = 600 | | | n = 1860 | | |
| 0–20 | 0.261 | 0.054 | 21.38 | 0.192 | 0.069 | 35.81 | 0.158 | 0.063 | 38.19 | 0.224 | 0.078 | 35.37 |
| 20–40 | 0.237 | 0.021 | 8.99 | 0.22 | 0.032 | 14.86 | 0.195 | 0.031 | 16.41 | 0.224 | 0.034 | 15.73 |
| 40–60 | 0.242 | 0.016 | 6.88 | 0.236 | 0.02 | 9.03 | 0.215 | 0.021 | 10.3 | 0.235 | 0.026 | 11.61 |
| 60–80 | 0.244 | 0.011 | 4.84 | 0.245 | 0.015 | 6.55 | 0.233 | 0.015 | 6.68 | 0.242 | 0.021 | 8.92 |
| 80–100 | 0.251 | 0.008 | 3.15 | 0.253 | 0.012 | 5 | 0.247 | 0.01 | 4.16 | 0.251 | 0.017 | 6.73 |
| 100–120 | 0.261 | 0.007 | 2.61 | 0.262 | 0.01 | 3.99 | 0.259 | 0.008 | 3.3 | 0.261 | 0.015 | 5.65 |
| 2003–2004 | n = 264 | | | n = 144 | | | n = 120 | | | n = 528 | | |
| 0–20 | 0.24 | 0.042 | 17.8 | 0.197 | 0.069 | 34.7 | 0.195 | 0.076 | 39.77 | 0.22 | 0.064 | 29.59 |
| 20–40 | 0.248 | 0.031 | 12.92 | 0.212 | 0.037 | 17.8 | 0.205 | 0.048 | 23.98 | 0.23 | 0.043 | 19 |
| 40–60 | 0.252 | 0.017 | 6.7 | 0.24 | 0.013 | 5.54 | 0.224 | 0.016 | 7.35 | 0.243 | 0.021 | 8.71 |
| 60–80 | 0.259 | 0.012 | 4.76 | 0.251 | 0.009 | 3.65 | 0.24 | 0.017 | 7.44 | 0.252 | 0.016 | 6.44 |
| 80–100 | 0.265 | 0.012 | 4.46 | 0.261 | 0.009 | 3.43 | 0.255 | 0.018 | 7.11 | 0.262 | 0.014 | 5.33 |
| 100–120 | 0.273 | 0.011 | 4.04 | 0.269 | 0.008 | 3 | 0.265 | 0.019 | 7.23 | 0.27 | 0.014 | 5.16 |
| 2004–2005 | n = 660 | | | n = 300 | | | n = 300 | | | n = 1260 | | |
| 0–20 | 0.262 | 0.035 | 13.33 | 0.19 | 0.07 | 36.87 | 0.18 | 0.078 | 42.15 | 0.226 | 0.075 | 33.49 |
| 20–40 | 0.243 | 0.027 | 11.24 | 0.225 | 0.033 | 14.73 | 0.16 | 0.03 | 18.65 | 0.219 | 0.048 | 22.41 |
| 40–60 | 0.251 | 0.017 | 6.92 | 0.237 | 0.013 | 5.47 | 0.166 | 0.026 | 16.42 | 0.227 | 0.041 | 18.52 |
| 60–80 | 0.249 | 0.013 | 5.16 | 0.237 | 0.008 | 3.52 | 0.182 | 0.03 | 17.65 | 0.23 | 0.034 | 15.25 |
| 80–100 | 0.25 | 0.008 | 3.36 | 0.244 | 0.005 | 2.2 | 0.203 | 0.032 | 17.4 | 0.237 | 0.027 | 11.64 |
| 100–120 | 0.243 | 0.007 | 2.89 | 0.24 | 0.004 | 1.86 | 0.209 | 0.029 | 14.96 | 0.234 | 0.021 | 9.56 |

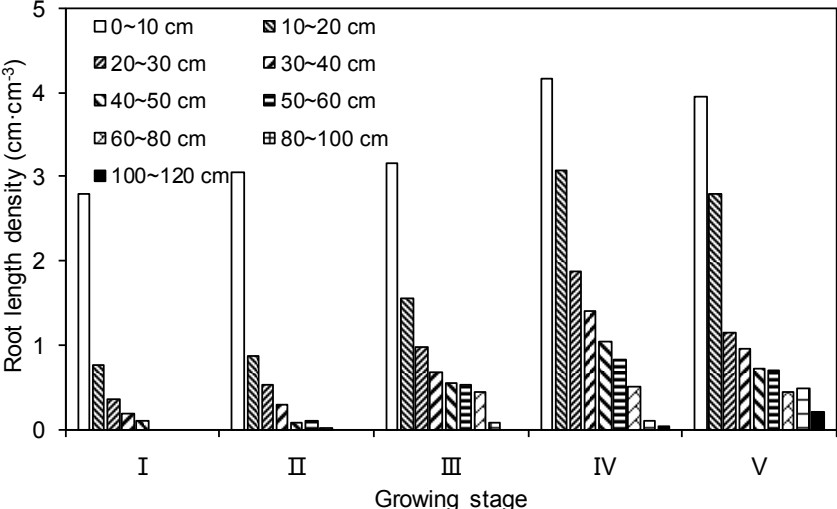

**Figure 3.** Density of winter wheat roots during different growth stages. Early: stage I; middle: stages II and III; last: stages IV and V.

Taking into account SD, CV of the soil water content at different layers in the soil profile, and the characteristics of growth and development of winter wheat roots, the appropriate SMS positioning in the soil should be at 0–40 cm, 0–60 cm, and 0–100 cm depths during the early stage, middle stage, and last stage, respectively.

### 3.2. Relationship of Soil Moisture at Different Layers in the Soil Profile

According to a correlation analysis of SWC at different layers in the soil profile (Table 2), significant linear correlations exist between the independent variables of SWC at neighboring layers. As a result, we can determine the corresponding value of SWC at neighboring layers based on one given layer.

**Table 2.** Correlation relationship of soil moisture variable at different depths in the soil profile (n = 3796).

| Depth | 10 cm | 30 cm | 50 cm | 70 cm | 90 cm | 110 cm |
|-------|-------|-------|-------|-------|-------|--------|
| 10 cm | 1 | | | | | |
| 30 cm | 0.9069 | 1 | | | | |
| 50 cm | 0.8856 | 0.9975 | 1 | | | |
| 70 cm | 0.8782 | 0.9959 | 0.9996 | 1 | | |
| 90 cm | 0.8751 | 0.9950 | 0.9992 | 0.9998 | 1 | |
| 110 cm | 0.8736 | 0.9946 | 0.9990 | 0.9997 | 0.9999 | 1 |

The value of the data in the table indicate the correlation coefficient for the variable of soil water content at different depths.

In order to select appropriate numbers of SWC monitoring points in the soil profile, a cluster analysis of the experimental data from five growing stages was performed. The cluster distance was calculated by the formula of Euclidean distance [24], and the results (Figure 4d) show that SWC at 50 and 70 cm soil depths and at 90 and 110 cm depths can be merged, since there were minimum distances of variables for SWC. Therefore, if we need to decrease the number of SMSs, we recommend maintaining them at 90 and 110 cm below the surface (because we can calculate the value of SWC at 110 cm using the value at 90 cm), or at 50 and 70 cm depths.

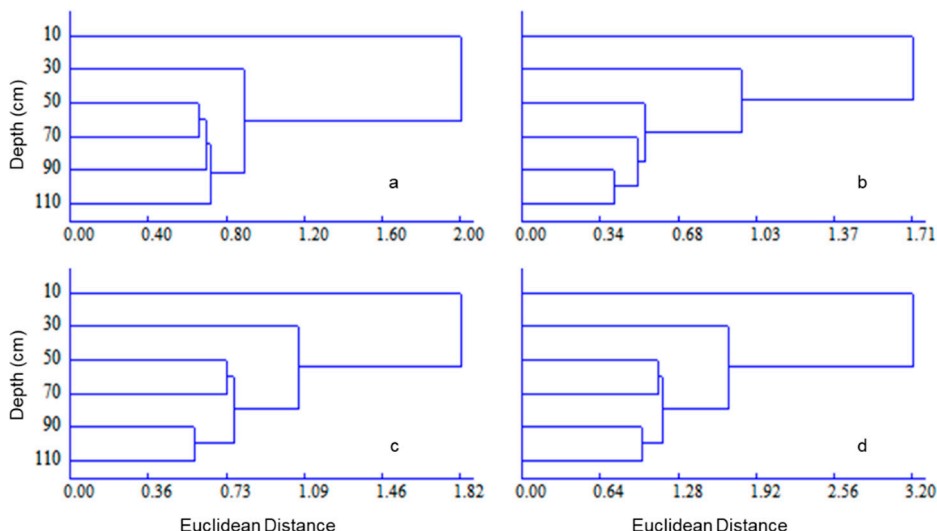

**Figure 4.** Cluster pedigree of soil moisture variables at different depths in the soil profile during different growing stages of winter wheat: (**a**) sowing to reviving stage; (**b**) returning green and jointing; (**c**) heading to maturity; and (**d**) whole growing season. X-axis is the distance of the variates for soil water content at different layers; y-axis is the different layers in the soil profile.

The description of variables for soil moisture was based on different classifications (Table 3). This can be explained as follows, during the whole growing season of winter wheat, the soil moisture

variables can be divided into two categories: variables of soil water content at 10 and 30 cm, representing shallow soil water conditions, and variables of soil water content at 50, 70, 90, and 110 cm, representing deep soil water conditions.

**Table 3.** Description of soil moisture variables at different depths in the soil profile. Number of classical refers to the categories divided for soil moisture variables in the soil profile; 2 refers to the two categories of soil moisture variables.

| Growing Stage | Depth (cm) | Number of Classical | | | | | Growing Stage | Depth (cm) | Number of Classical | | | | |
|---|---|---|---|---|---|---|---|---|---|---|---|---|---|
| | | 2 | 3 | 4 | 5 | 6 | | | 2 | 3 | 4 | 5 | 6 |
| Early stage | 10 | 1 | 1 | 1 | 1 | 1 | Last stage | 10 | 1 | 1 | 1 | 1 | 1 |
| | 30 | 1 | 2 | 2 | 2 | 2 | | 30 | 1 | 2 | 2 | 2 | 2 |
| | 50 | 2 | 3 | 3 | 3 | 3 | | 50 | 2 | 3 | 3 | 3 | 3 |
| | 70 | 2 | 3 | 3 | 3 | 4 | | 70 | 2 | 3 | 4 | 4 | 4 |
| | 90 | 2 | 3 | 3 | 4 | 5 | | 90 | 2 | 3 | 4 | 5 | 5 |
| | 110 | 2 | 3 | 4 | 5 | 6 | | 110 | 2 | 3 | 4 | 5 | 6 |
| Middle stage | 10 | 1 | 1 | 1 | 1 | 1 | Whole growing season | 10 | 1 | 1 | 1 | 1 | 1 |
| | 30 | 1 | 2 | 2 | 2 | 2 | | 30 | 1 | 2 | 2 | 2 | 2 |
| | 50 | 2 | 3 | 3 | 3 | 3 | | 50 | 2 | 3 | 3 | 3 | 3 |
| | 70 | 2 | 3 | 3 | 4 | 4 | | 70 | 2 | 3 | 3 | 4 | 4 |
| | 90 | 2 | 3 | 4 | 5 | 5 | | 90 | 2 | 3 | 4 | 5 | 5 |
| | 110 | 2 | 3 | 4 | 5 | 6 | | 110 | 2 | 3 | 4 | 5 | 6 |

Table 4 shows the AV, SD, and CV of SWC at different layers during the different growing stages. From Table 4 we can conclude that CV decreases with increased soil layer depth. CV for the variable of average SWC in the soil wetting layers is approximately equal to the variable of SWC at 30, 30, and 50 cm below the surface during the early, middle, and last stage of winter wheat, respectively. Figure 5 presents the relationship between soil moisture and variables of SWC at different layers in the soil profile. There is a significant correlation between average soil moisture in planned wetted layers and soil water content at 10 cm, 10 and 30 cm, and 30 and 50 cm during the early, middle, and last stage of winter wheat, respectively. $R^2$ is higher than 0.83, which indicates that average SWC could be obtained based on SWC at 10, 30, and 50 cm (Table 5). During the early and middle stages, we can calculate average SWC in soil wetting layers using the data observed at 10 cm below the surface with $R^2$ of 0.8876 and 0.8859; during the last stage, we can retrieve average SWC of the soil wetting layers using the data at 10 cm below the surface, but in this case, $R^2$ is only 0.5664. If $R^2$ is higher than 0.8000, at least two values of soil water content need to calculated.

**Table 4.** Average value, standard deviation, and coefficient of variation of soil water content at different layers during the different growing stages of winter wheat.

| Depth (cm) | Early Stage | | | Middle Stage | | | Last Stage | | |
|---|---|---|---|---|---|---|---|---|---|
| | AV ($cm^3 \cdot cm^{-3}$) | SD ($cm^3 \cdot cm^{-3}$) | CV (%) | AV ($cm^3 \cdot cm^{-3}$) | SD ($cm^3 \cdot cm^{-3}$) | CV (%) | AV ($cm^3 \cdot cm^{-3}$) | SD ($cm^3 \cdot cm^{-3}$) | CV (%) |
| | n = 1944 | | | n = 942 | | | n = 910 | | |
| 10 | 0.26 | 0.05 | 20.47 | 0.21 | 0.08 | 37.79 | 0.17 | 0.08 | 49.08 |
| 30 | 0.24 | 0.03 | 13.77 | 0.22 | 0.04 | 18.42 | 0.18 | 0.05 | 29.59 |
| 50 | | | | 0.24 | 0.03 | 12.01 | 0.20 | 0.05 | 24.21 |
| 70 | | | | | | | 0.22 | 0.05 | 22.12 |
| 90 | | | | | | | 0.23 | 0.05 | 20.02 |
| Planned wetting layer | 0.25 | 0.04 | 15.6 | 0.22 | 0.04 | 19.98 | 0.2 | 0.05 | 23.68 |

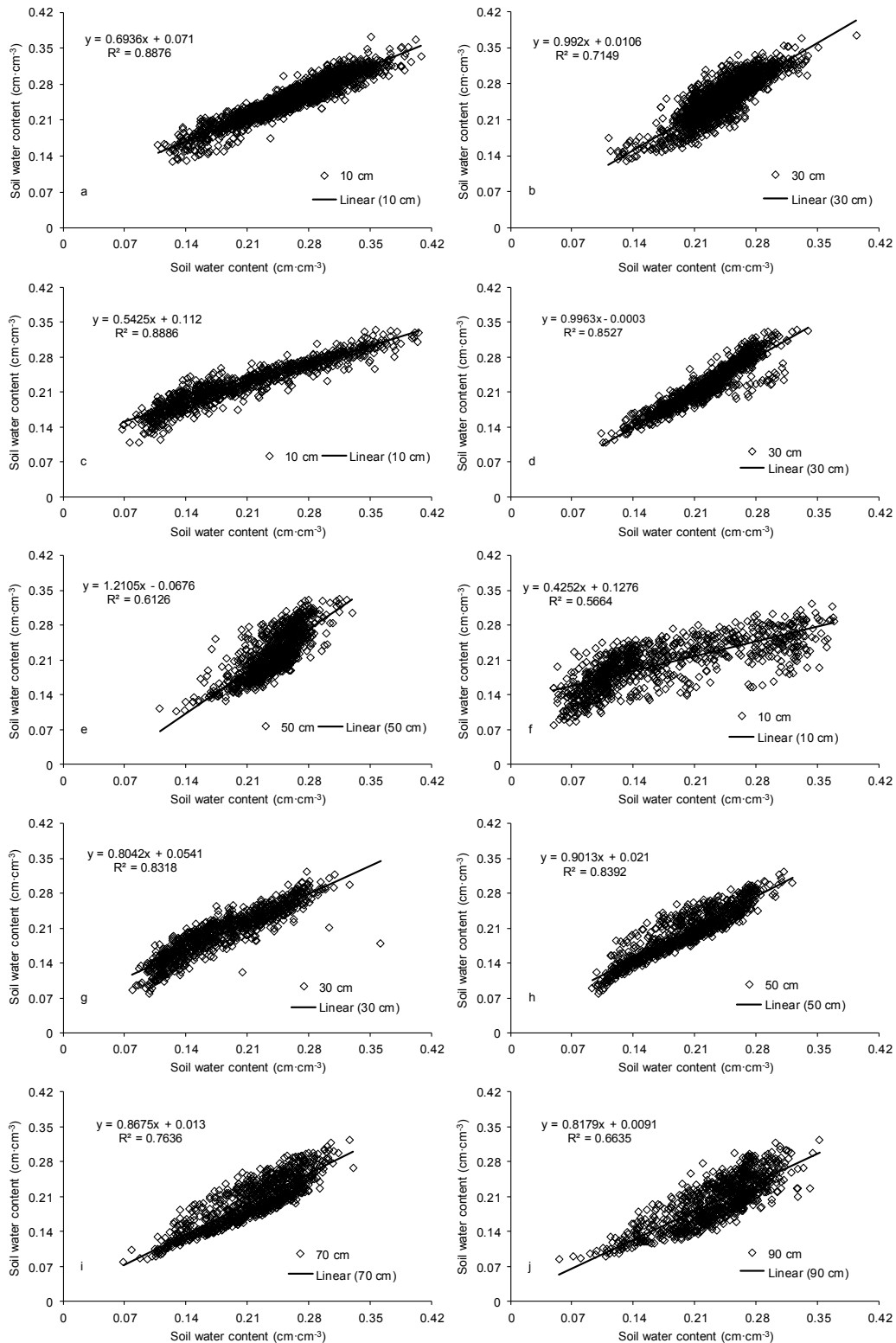

**Figure 5.** Relationship between average soil water content in planned wetted layers with soil water content at different depths in the soil profile during different growing stages of winter wheat: (**a**,**b**) early stage; (**c**–**e**) middle stage; and (**f**–**j**) last stage. *Y*-axis indicates average value of soil water content at 0–40 cm, *X*-axis indicates soil water content at 10 and 30 cm in (**a**) and (**b**), respectively. *Y*-axis indicates average value of soil water content at 0–60 cm, *X*-axis indicates soil water content at 10, 30, and 50 cm in (**c**), (**d**), and (**e**), respectively. *Y*-axis indicates average value of soil water content at 0–100 cm, *X*-axis indicates soil water content at 10, 30, 50, 70, and 90 cm in (**f**), (**g**), (**h**), (**i**), and (**j**), respectively.

**Table 5.** Regression equations for calculating soil moisture content in soil wetting layers based on the observed data of soil moisture with proper positioning of soil moisture sensors.

| Growing Stage | Number of Monitoring Points | Equations | $R^2$ |
|---|---|---|---|
| Early stage | 1 | $\theta_{0-40} = 0.693595\theta_{10} + 0.070977$ | 0.8876 |
| | 2 | $\theta_{0-40} = 0.5\theta_{10} + 0.5\theta_{30}$ | 1.0000 |
| Middle stage | 1 | $\theta_{0-60} = 0.542493\theta_{10} + 0.112039$ | 0.8859 |
| | 2 | $\theta_{0-60} = 0.326956\theta_{10} + 0.51998\theta_{30} + 0.039518$ | 0.9806 |
| | 3 | $\theta_{0-60} = 0.333333\theta_{10} + 0.333333\theta_{30} + 0.333333\theta_{50}$ | 1.0000 |
| Last stage | 1 | $\theta_{0-100} = 0.425212\theta_{10} + 0.127637$ | 0.5664 |
| | 2 | $\theta_{0-100} = 0.429791\theta_{10} + 0.503451\theta_{30} + 0.02198$ | 0.9135 |
| | 3 | $\theta_{0-100} = 0.208931\theta_{10} + 0.094185\theta_{30} + 0.64183\theta_{50} + 0.019467$ | 0.9750 |

Further analysis (Figure 6) shows relative errors (defined as absolute error divided by true value; absolute error is the difference between the magnitude of the true value and the observed one) between the variables of soil moisture in the soil wetting layers and the variables of SWC at different layers in the soil profile. For the early stage, there are similar relative errors between the average values of soil water content in planned wetted layers and the variables of soil water content at 10 and 30 cm below the surface, and the fraction of both samples with absolute relative errors less than 10.30% and 5.93% is 75% and 50%, respectively. For the middle stage, the fraction of 10 cm samples with absolute relative errors less than 26.38% and 15.73% is 75% and 50%, respectively. For the last stage, the fraction of 30 cm samples with absolute relative errors less than 19.16% and 9.93% is 75% and 50%, respectively.

Taking into account the CV of soil water content, the root distribution characteristics, the actual availability of different depths, and the operational sense of the monitoring of soil moisture in the field during the growing season of winter wheat, the suitable positioning of SMSs should be at 10, 30, and 50 cm below the surface in the soil profile.

*3.3. Validation of the Results*

The results of validation based on the observation data collected from the field experiment during the 2013–2014 winter wheat season are shown in Figure 6. The results indicate that there is excellent agreement between retrieved and observed SWC data. This indicates that with the calculated average SWC using the soil water content data observed at 10 cm depth during the early stage, the soil moisture in the planned wetted layer predicted by the formula (Table 5) was consistent with the observed data at 0–40 cm depth, with a root mean square error (RMSE) of 0.0254 $cm^3 \cdot cm^{-3}$ (n = 103, $R^2$ = 0.7112). During the whole growing season, the soil moisture in planned wetted layers predicted by the formula (Table 5) was consistent with the observed data, with RMSE of 0.0263 $cm^3 \cdot cm^{-3}$ (n = 247, $R^2$ = 0.7155). With calculated soil water moisture using soil water content data observed at 10 and 30 cm depth during the early stage, the soil moisture in planned wetted layers predicted by the formula (Table 5) was consistent with the observed data at 0–40 cm depth, with RMSE of 0.0000 $cm^3 \cdot cm^{-3}$ (n = 103, $R^2$ = 1.0000). During the middle stages, the soil moisture in planned wetted layers predicted by the formula (Table 5) was consistent with the observed data at 0–60 cm depth, with RMSE of 0.0174 $cm^3 \cdot cm^{-3}$ (n = 88, $R^2$ = 0.9069). During the last stage, the soil moisture in planned wetted layers predicted by the formula (Table 5) was consistent with the observed data at 0–100 cm depth, with RMSE of 0.0334 $cm^3 \cdot cm^{-3}$ (n = 56, $R^2$ = 0.8248). During the whole growing season, the soil moisture in planned wetted layers predicted by the formula (Table 5) was consistent with the observed data, with RMSE of 0.0190 $cm^3 \cdot cm^{-3}$ (n = 247, $R^2$ = 0.8749). With calculated soil water moisture using soil water content data observed at 10, 30, and 50 cm depths during the early stage, the soil moisture in planned wetted layers predicted by the formula (Table 5) was consistent with the observed data at 0–40 cm depth, with RMSE of 0.0000 $cm^3 \cdot cm^{-3}$ (n = 103, $R^2$ = 1.0000). During the middle stage, the soil

moisture in planned wetted layers predicted by the formula (Table 5) was consistent with the observed data at 0–60 cm depth, with RMSE of 0.0000 cm$^3$·cm$^{-3}$ (n = 88, R$^2$ = 1.0000). During the last stage, the soil moisture in planned wetted layers, predicted by the formula (Table 5), was consistent with the observed data at 0–100 cm depth, with RMSE of 0.0146 cm$^3$·cm$^{-3}$ (n = 56, R$^2$ = 1.0000). During the whole growing season, the soil moisture in planned wetted layers predicted by the formula (Table 5) was consistent with the observed data, with RMSE of 0.0069 cm$^3$·cm$^{-3}$ (n = 247, R$^2$ = 0.9773).

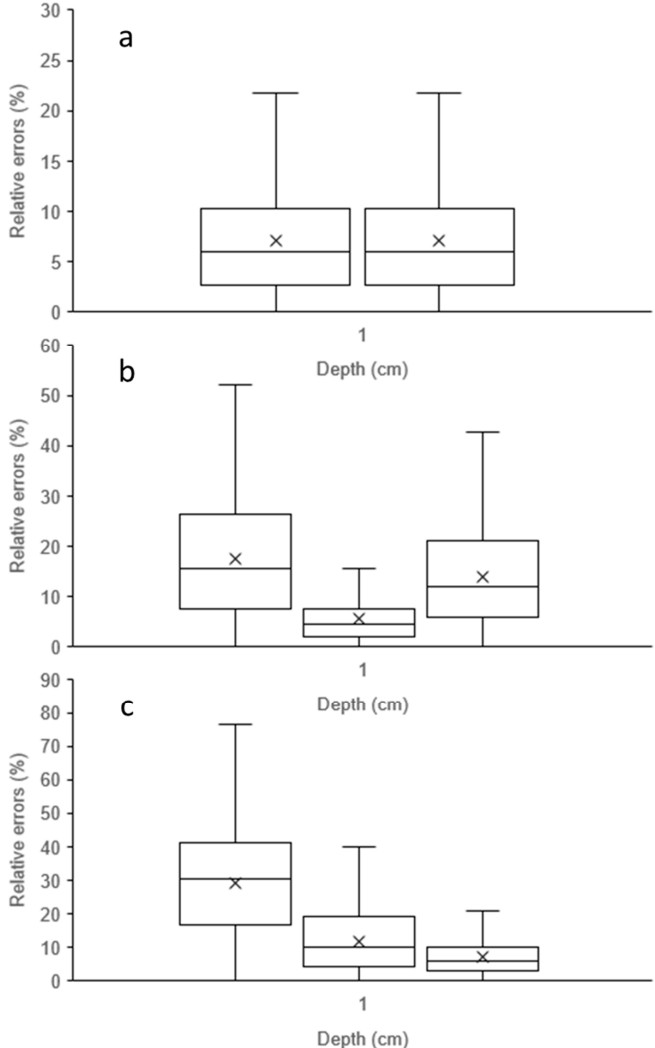

**Figure 6.** Box plots of relative errors of soil water content at different layers with average soil water content in planned wetted layers during different growing stages of winter wheat: (**a**) early stage; (**b**) middle stage; and (**c**) last stage. *Y*-axis indicates relative errors between the average value of soil water content at 0–40 cm and at different depths in the soil profile, *X*-axis indicates soil water content at 10 and 30 cm in (**a**). *Y*-axis indicates average value of soil water content at 0–60 cm, *X*-axis indicates soil water content at 10, 30, and 50 cm in (**b**). *Y*-axis indicates average value of soil water content at 0–100 cm, *X*-axis indicates soil water content at 10, 30, and 50 cm in (**c**).

Previous analysis shows that it is not suitable to use one observed value to represent the information of average soil moisture for soil wetting layers, especially during the last stage of winter wheat. In order to improve the precision, it is necessary to increase the number of observed points in the soil profile. Figure 7 suggests that it is possible to get quite close results of SWC in the simulated planned wetted zone based on two variables of soil water content at 10 and 30 cm below the surface, and three variables of soil water content at 10, 30, and 50 cm depth.

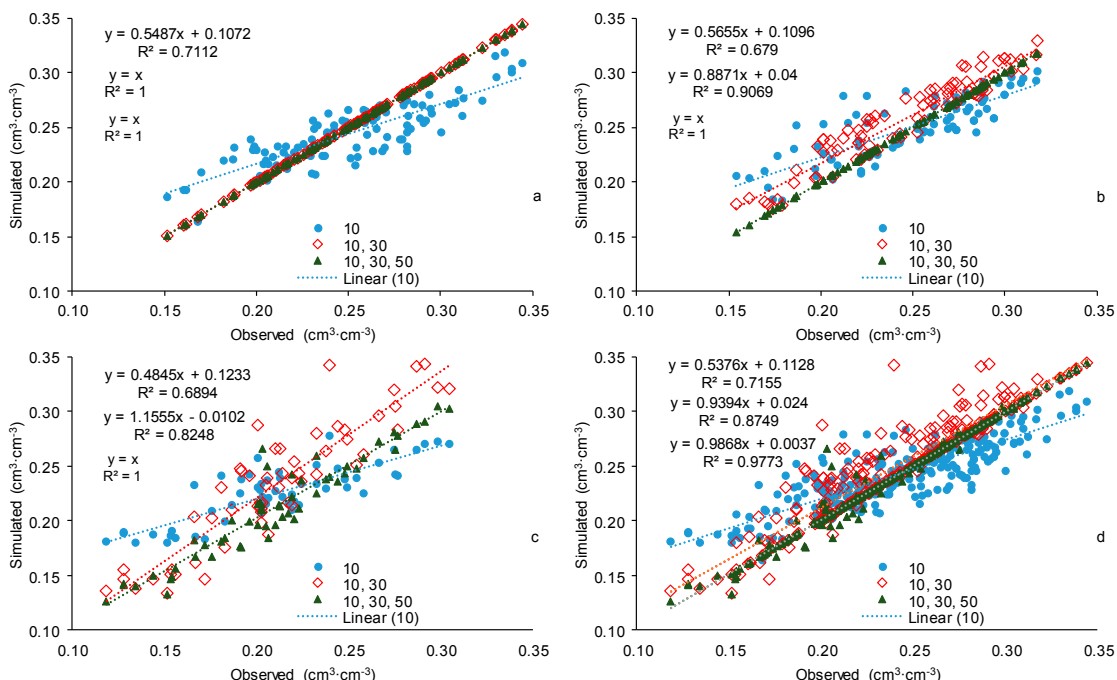

**Figure 7.** Validation results of simulated data with observed data: (**a**) early stage; (**b**) middle stage; (**c**) last stage; and (**d**) whole growing season of winter wheat.

### 3.4. Optimizing Irrigation Index of Winter Wheat with Border Irrigation

To determine water use efficiency (WUE), most researchers study the suitable irrigation index for winter wheat. Shen et al. and Meng et al recommended that suitable low irrigation limits were 50%, 65%, 70%, and 65% of field capacity (FC) during the sowing to reviving stage, returning green and jointing stages, heading to flowering stage, and filling to ripening stage, respectively; the suitable irrigation quota is 75 mm [25,26]. So we can determine the proper irrigation index based on the soil water content at proper positions of soil moisture sensors ($\theta_{10}$ and $\theta_{30}$) we recommend lower irrigation limits of 50% FC to apply irrigation during the early stage for winter wheat, 65% FC during the middle stage for winter wheat, 70% FC during heading to flowering stage (the last stage of winter wheat include heading to flowering stage and the ripening stage of the winter wheat), and 65% FC during the ripening stage (the last stage of winter wheat include heading to flowering stage and the ripening stage of the winter wheat) of winter wheat. The proper irrigation quota is 75 mm (Table 6).

**Table 6.** Optimizing irrigation limits based on the soil water content at different depths.

| Growing Stage | | Suitable Low Irrigation Limit | | Suitable Irrigation Quota (mm) |
|---|---|---|---|---|
| | | Based on SWC in Planned Wetted Layer | Based on SWC at Proper Positions of SMSs | |
| Early stage | Sowing to reviving stage | 50% FC | $(0.5\theta_{10} + 0.5\theta_{30}) \leq 50\%FC$ | 75 |
| Middle stage | Returning green and jointing stages | 65% FC | $(0.326956\theta_{10} + 0.51998\theta_{30} + 0.039518) \leq 65\%FC$ | 75 |
| Last stage | heading to Flowering stage | 70% FC | $(0.429791\theta_{10} + 0.503451\theta_{30} + 0.02198) \leq 70\%FC$ | 75 |
| | Filling to ripening stage | 60% FC | $(0.429791\theta_{10} + 0.503451\theta_{30} + 0.02198) \leq 65\%FC$ | 75 |

## 4. Discussion

Field soil moisture measurement is one of the most important factors in improving irrigation water management. Monitoring the real-time data of SWC not only ensures the quality and yield of agricultural crop, but also improves the irrigation efficiency. The number of SMSs and the locations where they are installed in the soil profile are fundamentally important in farmland soil moisture

monitoring. The sensors will be not able to detect changes in SWC if they are installed in regions where no roots exist. If these result are used to guide irrigation, it not only reduces the efficiency of irrigation water, but also affects the normal growth of crop, as a result reduce the yield and quality of the agriculture product. In contrast, if sensors are installed in soil zones with high root density, changes of SWC are easily reflected in the SMSs. Therefore, it is important to understand the impact of soil water distribution and consumption on crop growth, especially under irrigation.

The existing studies have focused on investigating the usage of different SMSs to measure soil water content [7,8,10]. Limited studies have been performed to investigate the suitable placement of moisture sensors in the soil profile based on water consumption at different crop growing stages and water uptake by roots from the soil [7,12,13,27]. However, these studies do not consider the root water absorption from the soil, and do not consider the different of the root distribution during the different growing stages for the crop, especially for the annual crops such as winter wheat, summer corn, and so on.

The results of this study show that the CV of SWC in the surface at 0–40, 0–60, and 0–100 cm fluctuates during the early (sowing to reviving), middle (returning green and jointing), and last (flowing and grain filling) stages of winter wheat, respectively. This is similar to the report by Gao et al. [17]. Our study also shows that most of the roots exist in the top layers of 0–40, 0–60, 0–60, and 0–100 cm during the sowing to reviving stage, returning green stage, jointing stage, flowering stage, and grain filling stage, respectively. This result is close to that reported by Zhang et al. [28]. In summary, we can conclude that planned wetted depths are 0–40, 0–60, and 0–100 cm from the surface during the early, middle, and last stages of winter wheat, respectively. Similar results have been reported in a previous study [26]. As result, there have great different of soil water storage at different depths to the crop growing during the different growing stages. In order to improve *WUE*, we should obtain the soil moisture in the root zoon (planned wetting layers) in the soil profile.

In this study, we obtained good simulation results to model average SWC using SWC at 10, 30, and 50 cm depths. These results are different from those in the study reported by Yang et al. [13]. They recommended that 10, 20, and 50 cm were suitable locations to install SMSs in a clover field. One of the main reasons for these differences may be that they only considered the relationship of the variables of soil water content at different layers in the soil profile, but not the different depths of the wetting layer during the different growing stages. In this study, we are emphatically analyze the relationship between the soil water content at depths and the soil moisture in the planned wetting layers during the different growing stages for the winter wheat, and then determined the proper number and the depths of the SMSs based on field experiment data (this data come from different field experiment sites and several growing seasons). Another reason could be that results based on experimental data collected from one treatment in one growing season have uncertainties different from the ones our experiments consider.

## 5. Conclusions

In conclusion, from this study, the main points obtained are as follows.

(1) Based on the analysis of soil water content and root distribution, the appropriate wetting layers using irrigation are 0–40 cm, 0–60 cm, and 0–100 cm during the early, middle, and last growth stages of winter wheat, respectively. This ensures crop growth and high irrigation water use efficiency.

(2) As a result of the appropriate wetting layers, in order to measure soil water content and schedule irrigation accordingly, the recommended depths of placements of soil moisture monitoring instruments are 0–40 cm, 0–60 cm, and 0–100 cm during the early, middle, and last stages, respectively.

(3) Observed SWC data at one depth in the soil profile is not representative enough to accurately calculate the average value of SWC for soil wetting layers. In order to accurately determine soil moisture content, we should investigate at least two observation points in the soil profile.

The linear regression model can be built using SWC at depths of 10 and 30 cm to predict average SWC in the soil profile. The results of validation showed that the developed model provided reliable estimates of average SWC in the planned wetting layer. In brief, this study suggests that the suitable positioning of soil moisture sensors is at depths of 10 and 30 cm below the soil surface.

**Author Contributions:** Conceptualization, J.Z. and Y.G.; Methodology, A.D. and X.S.; Validation, X.S. and J.L.; Data Analysis, X.S. and K.T.Z.; Writing-Original Draft Preparation, X.S. and K.T.Z; Writing-Review & Editing, X.S., K.T.Z., Y.L. and G.W.; Visualization, Z.M. and H.N.

**Funding:** This research was funded by the National Natural Science Foundation of China (No. 51309227), the Special Fund for Agro-Scientific Research in the Public Interest (201503130), and the China Agriculture Research System (CARS-3-1-30).

**Acknowledgments:** We thank Dr. Anque Guo for valuable comments on the draft of this manuscript.

**Conflicts of Interest:** The authors declare no conflict of interest.

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
