# Peer review of "Optimizing the Positioning of Soil Moisture Monitoring Sensors in Winter Wheat Fields"

_water, doi:10.3390/w10121707_

Reviewer 1 Report

The manuscript entitled "Optimizing the position of soil moisture monitoring sensor in Winter Wheat fields" describes a guideline for determine the optimal positioning as well as number of soil moisture sensor in an agricultural fields (winter wheat).

I found the manuscript well-written and scientific valuable. I think its publication can provide an important contribution for the scientific community. Therefore, I recommend its publication after minor revision. 

I recommend to submit the manuscript to the attention of a native speaker. Sometimes there are some typos and some nonnative language expressions which makes the manuscript hard to read.

2. I find the conclusions rather superficial. Try to summarise what you have done and to provide an answer to the scientific questions described in your abstract/ introduction

3. I assume that the methodology you described works in case of a flat terrain and I assume that the topography of your test area is rather gentle. I would expect that the linear relation between the soil moisture at the top and bottom layer would be not linear whenever your topography start to be relevant (I am thinking about hilly regions). Can you comment on that? Additionally, if my intuition is correct, I think this is a point which should be made clear in the manuscript.

Author Response

Thanks for the critical comments and constructive suggestions. Attachment is the detail.

Reviewer 2 Report

Major remarks

 I think that the your manuscript entitled  “Optimizing the Positioning of Soil Moisture

Monitoring Sensors in Winter Wheat Fields” brings new and valuable ideas and results to soil moisture monitoring in winter wheat fields and pedological studies on soil moisture.

This manuscript is quite systematic and thoughtful, having interesting results, analytical part as well as  experimental one (soil moisture measurement).  

This work is important from both scientific and practical point of view, and is suitable for publication in Water, after  minor revision.

1.     The conclusions are quite trivial and too short for such extensive and detailed study. Try to rewrite it putting here more questions you solved.

2.      In line 62 put with other works you mentioned the following one, you probably missed reading the literature: Comparative study of soil moisture estimations from SMOS satellite mission, GLDAS database, and cosmic-ray neutrons measurements at COSMOS station in Eastern Poland. M. Kędzior, J. Zawadzki. Geoderma, 2016.  283: 21-31.

This work describes results obtained using entirely new technique of soil moisture measurements, namely - cosmic-ray neutron measurements, and therefore ought to be included.

Minor remarks

 There is still small typos in the manuscript. I think a native speaker should read and correct it again.

 I recommend the manuscript for publication in Water, after above-mentioned minor revision.                                                                                                                                                                                                                                                  

Author Response

Thanks for the critical comments and constructive suggestions. Attachment is the detail.

Water EISSN 2073-4441 Published by MDPI AG, Basel, Switzerland RSS E-Mail Table of Contents Alert
Back to Top